# An Exogenous Ketone Ester Slows Tumor Progression in Murine Breast and Renal Cancer Models

**DOI:** 10.3390/cancers16193390

**Published:** 2024-10-04

**Authors:** Henry Nnaemeka Ogbonna, Zachary Roberts, Nicholas Godwin, Pia Muri, William J. Turbitt, Zoey N. Swalley, Francesca R. Dempsey, Holly R. Stephens, Jianqing Zhang, Eric P. Plaisance, Lyse A. Norian

**Affiliations:** 1Graduate Biomedical Sciences, Pathobiology, Pharmacology, and Physiology, University of Alabama at Birmingham, Birmingham, AL 35233, USA; hogbonna@uab.edu; 2Undergraduate Science and Technology Honors Program, University of Alabama at Birmingham, Birmingham, AL 35233, USA; zrob1598@uab.edu; 3Alabama School of Fine Arts, Birmingham, AL 35233, USA; ngodwin@sas.upenn.edu; 4Department of Nutrition Sciences, University of Alabama at Birmingham, Birmingham, AL 35233, USA; piamuri@uab.edu (P.M.); william.turbitt@emdserono.com (W.J.T.); 5Undergraduate Honors College, University of Alabama at Birmingham, Birmingham, AL 35233, USA; zswalley@uab.edu; 6Graduate Biomedical Sciences, Cancer Biology, University of Alabama at Birmingham, Birmingham, AL 35233, USA; fdempsey@uab.edu; 7Graduate Biomedical Sciences, Immunology, University of Alabama at Birmingham, Birmingham, AL 35233, USA; hollys@uab.edu; 8Department of Radiation Oncology, University of Alabama at Birmingham, Birmingham, AL 35233, USA; jianqing@uab.edu; 9Nutrition Obesity Research Center, University of Alabama at Birmingham, Birmingham, AL 35233, USA; 10O’Neal Comprehensive Cancer Center, University of Alabama at Birmingham, Birmingham, AL 35233, USA

**Keywords:** breast cancer, kidney cancer, ketone ester, diet

## Abstract

**Simple Summary:**

Exogenous ketone esters (KEs) are compounds that elevate blood ketones without dietary restriction and they show promise as anti-cancer agents. This study examined the effects of an exogenous ketone ester-supplemented (eKET) diet in mouse models of metastatic breast and kidney cancers. Our eKET diet slowed tumor growth and reduced metastatic spread to the lungs in both cancer types. The mechanisms behind these effects differed: the eKET diet downregulated genes involved in Wnt and TGFβ signaling in mammary cancers, whereas it downregulated genes related to hypoxia and DNA damage repair in tumor-bearing kidneys. The diet was safe for use overall but was associated with some negative physiological effects in mice with renal cancer. Thus, eKET diets could be an effective addition to current cancer treatments, but additional work is needed to fully understand their limitations.

**Abstract:**

Background/Objectives: Ketone esters (KEs) exhibit promise as anti-cancer agents but their impact on spontaneous metastases remains poorly understood. Although consumption of a ketogenic diet (KD) that is low in carbohydrates and high in fats can lead to KE production in vivo, the restrictive composition of KDs may diminish adherence in cancer patients. Methods: We investigated the effects of an exogenous ketone ester-supplemented (eKET), carbohydrate-replete diet on tumor growth, metastasis, and underlying mechanisms in orthotopic models of metastatic breast (4T1-Luc) and renal (Renca-Luc) carcinomas. Mice were randomized to diet after tumor challenge. Results: Administration of KEs did not alter tumor cell growth in vitro. However, in mice, our eKET diet increased circulating β-hydroxybutyrate and inhibited primary tumor growth and lung metastasis in both models. Body composition analysis illustrated the overall safety of eKET diet use, although it was associated with a loss of fat mass in mice with renal tumors. Immunogenetic profiling revealed divergent intratumoral eKET-related changes by tumor type. In mammary tumors, Wnt and TGFβ pathways were downregulated, whereas in renal tumors, genes related to hypoxia and DNA damage repair were downregulated. Conclusions: Thus, our eKET diet exerts potent antitumor and antimetastatic effects in both breast and renal cancer models, albeit with different modes of action and physiologic effects. Its potential as an adjuvant dietary approach for patients with diverse cancer types should be explored further.

## 1. Introduction

The use of dietary interventions to slow tumor progression and/or enhance therapeutic efficacy is an active area of research that has led to promising results in both patients and pre-clinical models reviewed in [1,2]. Often, these interventions rely upon glucose restriction, due to the well-documented reliance of many cancer cells on glucose metabolism to sustain their continuous proliferation and energetic needs. Ketogenic diets (KDs), which typically have a carbohydrate content of 5–10% with the remainder of calories coming predominantly from fat rather than protein, have been evaluated for their ability to slow tumor progression, with promising but mixed results [3,4,5]. Compounding this fact are the palatability issues with KDs that can make adherence challenging [4,5], particularly for cancer patients who frequently experience decreased appetite and nausea. KDs increase fatty acid oxidation in the liver, which leads to the production and secretion of ketones such as β-hydroxybutyrate (βHB) and acetoacetate (AcAc). Ketones have numerous, previously reported intrinsic anti-cancer properties: they reduce systemic glucose availability, inhibit histone deacetylases [6], reduce inflammation [7,8], impair matrix metalloproteinase activity [9], and may enhance T cell immunity [10]. For these reasons, there has been an increasing interest in using ketone-enhancing diets as anti-cancer therapeutics.

To bypass the palatability and adherence issues related to KDs, we and others have begun to evaluate the effects of providing exogenous ketone esters (KE) in a diet that contains standard levels of carbohydrates, fat, and protein. We call this the “exogenous ketone ester” (eKET) diet. Numerous studies have demonstrated the safety of consuming exogenous KEs, as well as their wide-ranging general health and metabolic benefits [11,12,13,14,15,16]. The use of exogenous KEs in the cancer setting has received growing attention. An early paper by D’Agostino and colleagues showed that a diet supplemented with exogenous KEs reduced the viability of a glioblastoma cell line in vitro and prolonged survival in tumor-challenged mice [17]. Ferrere et al. reported that in a non-metastatic melanoma model, the use of either a KD or exogenous βHB slowed primary tumor growth and enhanced the efficacy of a combinatorial anti-CTLA-4/anti-PD-1 immunotherapy [18]. A 2024 study by Murphy et al. found that exogenous βHB showed efficacy in a model of murine prostate cancer, in terms of inhibiting histone deacetylases and augmenting anti-PD-1 immune checkpoint blockade, without inducing the detrimental weight loss seen in mice on a full KD [6]. Although these results show the exciting potential for exogenous KEs to slow tumor growth and boost immunotherapeutic efficacy, knowledge about the ability of exogenous KEs to act as anti-cancer agents in the context of spontaneously metastasizing orthotopic cancers remains poorly understood.

Here, we evaluated the effects of an exogenous KE to slow tumor growth in vitro and in vivo, using pre-clinical models of orthotopic, spontaneously metastasizing breast and renal cancers. We intentionally selected these models based on a previous study by Qian et al. whose analysis of over 10,000 tumor specimens from The Cancer Genome Atlas (TCGA) revealed that both breast cancer and renal carcinoma exhibit high glycolytic activity and reduced ketone ester metabolism [19]. Thus, we evaluated the safety and efficacy of our eKET diet, with a particular focus on whether this dietary approach—in the absence of any therapy—could diminish both primary tumor growth and metastatic lesions in the lung. Our results suggest that the use of the eKET diet could be translated into clinical use as a means to slow tumor progression, although additional work in this area remains to be completed.

## 2. Materials and Methods

### 2.1. Animals and Diets

Wild-type BALB/c female mice (6–8 weeks old) were obtained from Charles River Laboratories (NCI-Frederick colony). All mice were housed in specific-pathogen-free conditions before use and were maintained at room temperature on a 12–12-h light–dark cycle. After arrival, mice were acclimated for one week on a control diet (CON, Dyets, Inc., #104419, Bethlehem, PA, USA), comprising 20% kcal protein, 17% kcal fat, and 63% kcal carbohydrates. Subsequently, mice were challenged with either Renca-Luc cells or 4T1-Luc cells (see below). Immediately following the tumor challenge, mice were randomized to either continue on the CON diet or transition to an isocaloric exogenous ketone ester (eKET, Dyets, Inc., #104424, Bethlehem, PA, USA) diet. The eKET diet mirrored the CON diet’s macronutrient composition, with the exception of 25% carbohydrate kcal being replaced by R,S-1,3-butanediol acetoacetate (BD-AcAc_2_, Disruptive Enterprises, Durham, NC, USA), a ketone diester and upstream in vivo precursor of β-hydroxybutyrate (βHB) and acetoacetate (AcAc). Therefore, the eKET Diet is a carbohydrate-replete diet with normal percentages of fat and protein, which distinguishes it from “ketogenic” diets. All animal experiments were conducted in accordance with the guidelines and approval of the Animal Resources Program and Institutional Animal Care and Use Committee at the University of Alabama at Birmingham (UAB) under protocol #IACUC 20177 (LAN).

### 2.2. Cell Culture

The 4T1 murine mammary carcinoma cell line (ATCC), syngeneic to BALB/c mice, was cultured as per the supplier’s instructions and engineered to stably express firefly luciferase (4T1-Luc) via lentiviral transduction as previously described [20]. Selective pressure for transduced cells was achieved via culture with 1 mg/mL puromycin. The Renca renal carcinoma cell line (ATCC), also syngeneic to BALB/c mice, was cultured as per the supplier’s instructions and engineered to stably express firefly luciferase (Renca-Luc) via lentiviral transduction, as previously described [21]. Selective pressure for transduced cells was achieved via culture with 1 mg/mL puromycin. Before use, both cell lines were found to be free from mycoplasma contamination or pathogens that may commonly infect cell lines.

### 2.3. Seahorse Assay

Cultured 4T1-Luc and Renca-Luc cells were harvested, counted, and plated in a Seahorse assay plate at 40,000 cells/well. After 24 h, culture media was replaced with phenol-free DMEM supplemented with sodium pyruvate, L-glutamine, and glucose, and the plate was placed in a non-CO_2_ incubator for one hour. Seahorse cartridges were loaded with modulators of respiration, including oligomycin, carbonyl cyanide-4 (trifluoromethoxy) phenylhydrazone (FCCP), and rotenone and antimycin and the Seahorse XF Cell Mito Stress Test was performed using the Agilent XF24 Seahorse Bioanalyzer per manufacturer’s instructions. Oxygen consumption rate (OCR) and extracellular acidification rate (ECAR) were measured at designated intervals and used to calculate parameters of metabolic function: ATP-linked respiration (basal OCR—oligomycin-induced OCR reduction), maximal respiration (OCR value post-FCCP injection—OCR valley post rotenone and antimycin injection), and spare respiration (maximal respiration—basal respiration).

### 2.4. In Vivo Tumor Modeling

To model metastatic renal cancer, mycoplasma-free Renca-Luc cells (5 × 10^4^ in 100 μL volume/mouse) were orthotopically injected into the left kidneys of 6–8-week-old female BALB/c mice as previously described [22,23,24]. Following intraperitoneal injection of 1 mg/mL D-luciferin (GoldBioChem, St. Louis, MO, USA), the growth of primary renal tumors was monitored by bioluminescent imaging (BLI) using an IVIS Lumina III imager (Perkin Elmer, Waltham, MA, USA) at the UAB Small Animal Imaging Facility. At endpoint, primary renal tumors were excised and weighed; lung metastases were assessed by BLI. To model metastatic breast cancer, mycoplasma-free 4T1-Luc cells (1 × 10^6^ in 200 μL volume/mouse) were injected into mammary fat pad #9 of 6–8-week-old female BALB/c mice as per our prior protocols [20,25]. Tumor growth was monitored by caliper measurements. At the experimental endpoint, primary mammary tumors were excised and weighed; lung metastases were quantitated by BLI.

### 2.5. Blood Glucose and B-Hydroxybutyrate Measurements

To assess the impact of the dietary interventions on glucose homeostasis and ketone body metabolism, a fasting/refeeding protocol was implemented on Day 16 post-tumor challenge. Animals were fasted for six hours prior to an intraperitoneal glucose tolerance test (GTT). Baseline blood glucose levels were measured (time 0), followed by intraperitoneal administration of a 2 g/kg glucose bolus. Blood glucose levels were then measured over the next 120 min using a handheld combined glucometer/ketone meter (KetoMojo, Napa, CA, USA). Because point-of-care ketone device technology is not capable of measuring circulating acetoacetate concentrations, only concurrent blood β-hydroxybutyrate (β-HB) concentrations were measured.

### 2.6. Quantitative Magnetic Resonance

Quantitative magnetic resonance (QMR) was employed to investigate the effects of the exogenous ketone ester (eKET) diet on body composition in tumor-challenged mice. Using an EchoMRI™ 3-in-1 system (Echo Medical Systems, Houston, TX, USA), body mass, fat mass (FM), and lean mass (LM) were measured at two timepoints: Day-1 pre-tumor challenge (all mice on CON diet) and Day 20 post-tumor challenge (mice on CON or eKET diets). Measurements were performed in collaboration with the UAB Small Animal Phenotyping Core, following previously described procedures [26]. On Day 21, primary tumors were removed, weighed, and then subjected to ex vivo QMR. Values for tumor weight, fat mass, and lean mass were subtracted from Day 20 data on intact mice to achieve calculated values for tumor-free body weight, fat mass, and lean mass.

### 2.7. NanoString Immunogenetic Profiling

Whole tumors were harvested from 4T1-Luc and Renca-Luc tumor-bearing mice maintained on either CON or eKET diets at Day 21 post-tumor challenge. Approximately 0.2 g of tumor tissue was submerged in 0.5 mL RNALater (Ambion, Waltham, MA, USA) and then stored at −80 °C. Total RNA was batch processed from homogenized tumor samples (Zymo Research, Irvine, CA, USA) then concentration and purity were assessed using a Take3 micro-volume plate and Synergy reader (Agilent, Santa Clara, CA, USA). Gene expression profiling was performed using the nCounter Mouse PanCancer Immune Profiling Panel (NanoString Technologies, Seattle, WA, USA). Raw gene expression data were processed and normalized using NanoString’s nSolver Analysis Software v4.0, and gene expression analyses were performed using the Rosalind analysis platform. Differentially expressed genes (DEGs) were identified as those with a *p*-value < 0.05 and fold change >±1.5. Due to the exploratory nature of our study and intent for validation, we first used raw *p* values to identify DEGs. In the 4T1 model, these steps yielded >120 DEGs, so we took the additional step of reporting DEGs in this model only when the *p*-adjusted value was <0.05, to account for false discovery rate (FDR) and multiple comparisons. The complete list of differentially expressed genes (DEGs) for each model is provided in Appendix A.

### 2.8. Western Blot

Cells were harvested and lysed in RIPA buffer supplemented with a protease and phosphatase inhibitor cocktail. The resulting protein extract was stored at −80 °C until use. Proteins were separated by SDS-PAGE on a 10% polyacrylamide gel using a constant voltage of 100 V until complete resolution. Subsequently, proteins were transferred to a nitrocellulose membrane that had been pre-soaked in a transfer buffer. The membrane was blocked with milk for 1 h then incubated with primary antibodies against H3K9Ac (cat. no.: 9649T, Cell signal technology, Danvers, MA, USA) and β-actin (as loading control; cat. no NB600-501, Novus biologicals, Centennial, CO, USA). Following removal of the primary antibodies, the membrane was incubated with horseradish peroxidase (HRP)-conjugated secondary antibodies against mouse and rabbit IgG (1:1000 dilution) for 2 h at room temperature in the presence of bovine serum albumin (BSA) and protected from light. After three washes with Tris-buffered saline with Tween20 (TBST), immunoreactive bands were visualized using a chemiluminescent imager.

### 2.9. Statistical Analyses

Statistical analyses were conducted with GraphPad Prism (Version 10.1.2). Values are presented as means ± SEM. Normality was evaluated through the Shapiro–Wilk test. For comparisons involving four groups, either the non-parametric Kruskal–Wallis test or a one-way ANOVA (parametric) was employed, followed by appropriate post-hoc pairwise comparisons: Mann–Whitney U test (non-parametric), unpaired two-tailed *t*-test (parametric), or Tukey’s multiple comparisons tests. In studies assessing repeated measures across multiple time points and involving more than two groups, a two-way repeated-measures ANOVA was implemented. Comparisons among the three groups utilized either the non-parametric Kruskal–Wallis test with Dunn’s multiple comparisons test or a one-way ANOVA (parametric) with Tukey’s or Sidak’s multiple comparisons test, as dictated by the data distribution. For comparisons between two groups, a student’s *t*-test or Mann–Whitney test was used, as appropriate. Throughout, statistical significance is indicated as follows: * *p* < 0.05, ** *p* < 0.01, *** *p* < 0.001, **** *p* < 0.0001.

## 3. Results

### 3.1. Metabolic Phenotyping Reveals Divergent Bioenergetic Profiles between 4T1 Mammary and Renca Renal Carcinoma Cell Lines

Cancer cells adapt to dynamic stressors and proliferative needs by reprogramming their metabolism to fuel their dysregulated division [27]. However, specific cancer cell types exhibit marked differences in their basal glucose and oxygen consumption rates, reflecting varying metabolic dependencies for growth and proliferation [28]. To evaluate the effects of our eKET diet in models of spontaneously metastasizing cancers, we selected cell lines that represent cancer types known to be highly glycolytic and that metastasize following orthotopic implantation in syngeneic mice. To begin, we assessed potential metabolic differences in our 4T1-Luc (mammary carcinoma) and Renca-Luc (renal carcinoma) cells. Seahorse assays were used to measure basal oxygen consumption rate (OCR) and extracellular acidification rate (ECAR)—indicators of mitochondrial respiration and glycolytic activity, respectively. Our analysis revealed distinct metabolic profiles between the two cell lines (Figure 1). The 4T1-Luc cells exhibited a higher oxygen consumption rate (OCR) at baseline and following oligomycin-induced ATP synthase inhibition (Figure 1A), indicating elevated mitochondrial and non-mitochondrial oxygen consumption. Quantification of key metabolic parameters further highlighted the metabolic divergence between the two cell lines. Basal and ATP-linked respiration was significantly higher in 4T1-Luc cells compared to Renca-Luc cells, consistent with their elevated OCR (Figure 1B). Although maximal respiration and proton leak tended to be higher in 4T1-Luc cells, these differences did not reach statistical significance. Renca-Luc cells demonstrated a slightly higher spare respiratory capacity, suggesting a modest ability to respond to increased energy demands compared to 4T1-Luc cells. Analysis of the ECAR revealed no significant difference in basal glycolytic activity between the two cell lines, although 4T1-Luc cells exhibited a significantly higher glycolytic peak ECAR, indicating a greater capacity for glycolytic flux (Figure 1C). These results demonstrate that 4T1-Luc cells exhibit a more robust metabolic phenotype than Renca-Luc cells, as characterized by elevated mitochondrial respiration and glycolytic flux, which suggests that 4T-1 cells may be more susceptible to growth regulation by ketones.

### 3.2. Ketones Do Not Impair 4T1-Luc or Renca-Luc Growth or Histone Deacetylase Activity In Vitro

Next, we assessed the impact of ketones on cancer cell growth and viability in vitro, as prior reports had indicated the direct effects of ketones on tumor cell proliferation [6,17]. Cell growth was assessed by measuring total cell counts at 72 h post-treatment, whereas viability was evaluated by quantifying live cell counts. Cells were cultured in medium alone or with 5 mM β-hydroxybutyrate (R-βHB), 5 mM acetoacetate (AcAc), or a combination of 2.5 mM β-HB and 2.5 mM AcAc. In contrast to reports on other cell lines, we found no significant inhibition of tumor cell proliferation or viability after culturing 4T1 or Renca cells with R-βHB or AcAc (Figure 1D,E).

Prior studies indicated that βHB possesses intrinsic histone deacetylase inhibitor (HDACi) activity [6], which modulates gene expression and could influence tumor growth in vivo. To investigate this possibility, we examined the effects of ketone administration (R-βHB and AcAc, singly or in combination) on cell cultures and included as a control the established HDACi Vorinostat to permit evaluation of histone acetylation in 4T1-Luc and Renca-Luc cells. Consistent with its known HDACi activity, Vorinostat treatment significantly increased levels of acetylated histone H3 lysine 9 (H3K9Ac), a marker of chromatin relaxation and transcriptional activation (Figure 1F,G). Notably, R-βHB and AcAc induced only trending increases in H3K9Ac levels. This effect was not amplified when cells were treated with a combination of both ketones, suggesting no synergistic inhibition of HDACs in either cell line.

### 3.3. The eKET Diet Displays Efficacy in Slowing Tumor Progression in Mammary and Renal Carcinoma Models

The anti-cancer effects of exogenous KEs in mice with spontaneously metastasizing tumors remain understudied. Two recent reports examined the ability of exogenous KEs to slow primary tumor growth [6,18], but neither evaluated effects on spontaneous metastases, which are essential for understanding the clinical implications of exogenous KE administration. Despite the inability of βHB and AcAc to directly impair 4T1-Luc or Renca-Luc growth in vitro, we investigated the effects of our eKET diet on orthotopic tumor progression and metastasis in mice. Prior to tumor implantation (Day 7), mice were started on the control diet (CON) (Figure 2A). Immediately following tumor implantation (Day 0), mice were randomized to either continue on the CON diet or switch to the eKET diet (Figure 2A,B), a carbohydrate-replete diet wherein 25% of the carbohydrate-derived calories were replaced with BD-AcAc_2_ (Figure 2A), a ketone diester precursor to R-β-HB and AcAc. This formulation was selected as we have previously found that it does not significantly impact food intake or body weight in tumor-free mice [29,30]. Analysis of blood R-βHB concentrations in mice following a six-hour fast and two-hour re-feed confirmed a significant elevation in circulating R-βHB in both tumor models at day 16 post-tumor challenge for mice on the eKET diet (Figure 2C,G). The observed blood R-β-HB concentrations in eKET diet mice were 0.98 ± 0.08 mmol/L (Mean ± SEM) for Renca-Luc mice and 1.04 ± 0.332 mmol/L (Mean ± SEM) for 4T1-Luc mice. These levels were consistent with those reported in humans in a mild, healthy state of nutritional ketosis [31,32]. Simultaneous fasted blood glucose concentrations revealed no significant alterations between mice on CON versus eKET diets (Appendix A). However, glucose tolerance testing in fasted mice revealed divergent responses between the two tumor models. The 4T1-Luc-bearing mice on the eKET diet displayed significantly higher circulating glucose levels than the mice on control diet over the duration of the test period, whereas Renca-Luc-bearing mice on the eKET diet showed a diminished capacity to regulate blood glucose and displayed lower overall area under the curve values compared to the mice on control diet (Appendix A). Thus, mice with 4T-1 tumors on the eKET diet entered a healthy state of ketosis without the systemic perturbations in glucose regulation that were observed in mice with orthotopic Renca-LUC tumors.

Our evaluation of primary tumor growth and metastasis revealed eKET diet efficacy in both tumor models. Longitudinal monitoring of 4T1-Luc tumor outgrowth via calipers demonstrated a significant reduction in mammary tumor outgrowth with eKET diet (Figure 2D), as did endpoint analysis of excised tumor weights at Day 21 (Figure 2E). Given the propensity of 4T1-Luc tumors to metastasize spontaneously to the lungs, we also investigated lung tumor burdens. Quantification of spontaneous lung metastases by BLI of excised lungs revealed a significant, 47% reduction in lung tumor burdens (Figure 2F), illustrating a potent antimetastatic effect of the eKET diet. Similar findings were made in the Renca-Luc model. Here, analysis of Renca-Luc renal tumors by BLI revealed that the eKET diet significantly impeded tumor outgrowth over time, as evidenced by reduced BLI values (Figure 2G), resulting in significantly smaller excised tumor weights at the experimental endpoint (Figure 2H). Although lung tumor burdens at Day 21 were not significantly reduced, the mean BLI value from Renca-Luc mice on the eKET diet reflected a 96% reduction in metastatic lung tumors (Figure 2I) versus mice on the CON diet. Importantly, initial tumor BLI signals were comparable between CON and eKET diet groups for both tumor models at day 7 post-implantation, indicating that the eKET diet did not alter tumor establishment but rather exerted its effects on subsequent progression (Appendix A). Collectively, these findings demonstrate that our eKET diet effectively inhibits the outgrowth of both renal and mammary tumors in vivo, and importantly, mitigates their metastatic spread to the lungs.

### 3.4. The Safety Profile of eKET Diet Administration Differs between Mammary and Renal Tumor Models

Previous studies have examined the effects of KE administration on body weight and body composition in non-tumor models and found it to be safe for use [29,30,33]. In agreement with these prior reports, we found that in tumor-free mice, eKET administration did not significantly alter excised calf muscle, liver, spleen, or contralateral kidney weights after three weeks of diet administration, indicating that the diet is well-tolerated and does not induce adverse systemic effects in healthy animals (Appendix A). However, it was critical for us to evaluate eKET diet safety in tumor-bearing mice, as growing solid tumors exert large metabolic demands [34], and it was possible that given the cancer dissemination present in our models, we might find undesirable side effects in tumor-challenged mice on the eKET diet. Thus, we examined endpoint organ weights in tumor-free and tumor-bearing mice that had been randomized to CON or eKET diets. In 4T1-Luc mammary tumor-bearing mice, the eKET diet similarly had no significant effect on the endpoint (Day 21 post-challenge) calf muscle, white gonadal fat, contralateral kidney, or liver weights (Figure 3A). As expected for this aggressive model [35], splenomegaly was observed and the eKET diet significantly reduced spleen weights in these mice, suggesting a systemic effect of the diet on immune responses (Figure 3A).

Mice with Renca-Luc tumors exhibited divergent responses to the eKET diet. Although no changes were observed in the calf muscle, gonadal fat, or contralateral kidney weights, Renca-Luc mice on the eKET diet had significantly reduced liver and spleen weights (Figure 3B), the latter in a model not known to induce splenomegaly post-tumor challenge (see Appendix A).

To more definitively assess the effects of our eKET diet on the body composition of tumor-bearing mice, quantitative magnetic resonance (qMR) analyses were performed on individual animals at both pre-tumor-implantation (Day 1) and post-tumor-implantation (Day 20) time points. Excised tumor composition was evaluated on Day 21 and subtracted from Day 20 data, to permit evaluation of animal body composition without the confounding effects of solid tumor masses. This analysis revealed distinct, tumor-specific effects of the eKET diet on body composition. Although body weights were not significantly different across experimental groups post-tumor challenge, mice with 4T1-Luc tumors on the eKET diet experienced a trending increase in tumor-free body weights relative to mice on the CON diet (Figure 4A), whereas mice with Renca-Luc tumors on eKET diet displayed a non-significant trend toward weight loss (Figure 4D). In mice with 4T1 tumors, eKET diet feeding led to significantly increased fat mass post-tumor challenge relative to mice on the CON diet (Figure 4B), without altering the trend in declining lean mass that was observed in CON mice from Day 1 to Day 20 (Figure 4C). In contrast, mice with Renca-Luc tumors on the eKET diet experienced a significant reduction in fat mass at Day 20 relative to mice on the CON diet, a trend not observed with lean mass (Figure 4E,F). Of note, although food intake decreased slightly in both tumor models at day 13 post-tumor challenge, this occurred equivalently for mice on CON and eKET diets (Appendix A). Thus, reduced food intake did not appear to be the cause of the trending body weight reductions or loss of fat mass observed specifically in mice with Renca-Luc tumors. Collectively, these data indicate that the use of our eKET diet in mice appears to be more readily tolerated in mice with mammary tumors, as they did not exhibit the reductions in liver weights or fat mass seen in mice with renal tumors.

### 3.5. Immunogenetic Profiling Reveals Distinct Tumor-Specific Mechanisms Underlying the Antitumor and Antimetastatic Effects of the eKET Diet

To begin elucidating mechanisms underlying the observed inhibitory effects of the eKET diet on tumor growth and metastasis, we conducted transcriptomic profiling via NanoString analysis of RNA from 4T1-Luc or Renca-Luc tumors derived from mice fed either CON or eKET diets. The results revealed distinct and tumor-specific alterations in gene expression patterns, providing insights into the distinct pathways modulated by the eKET diet in mice with mammary versus renal tumors. In 4T1-Luc tumors, the use of the eKET diet caused a substantial remodeling of the tumor microenvironment, with more than 127 differentially expressed genes (DEGs) identified that exceeded fold change and unadjusted *p*-value thresholds. Examining only those 23 DEGs that showed a significant *p*-adjusted value (Appendix A) revealed a massive reduction in gene expression, including genes involved in cell adhesion (*Ncam1*), growth factor signaling (*Wnt4*, *Wnt10A*, *Fgr1*), and metastasis-associated pathways (*Zeb1*) (Figure 5A). The sole upregulated gene was CCL26. Pathway analysis showed down-regulation of Hedgehog, Wnt, and TGFβ signaling.

In mice with Renca-Luc tumors, eKET diet use was associated with distinct patterns of both up- and down-regulated genes, none of which reached a significant *p*-adjusted value, illustrating the more modest tumor microenvironment remodeling present in these animals. We therefore evaluated those genes (*n* = 22) that exceeded fold change and unadjusted *p*-value thresholds (Appendix A), to gain mechanistic insight into interactions between the eKET diet and renal tumors. Key downregulated genes include those involved in angiogenesis (*Adm*), hypoxia signaling (*Hif1a*), DNA repair (*Fanca*), and cell cycle regulation (*Wdr76*); upregulated genes included those associated with inflammation (*Ptgs2*) and cell adhesion (*Itga2*) (Figure 5B). Thus, our findings reveal that the eKET diet exerts potent antitumor and antimetastatic effects through tumor-specific mechanisms in mammary versus renal tumors, highlighting its potential as a versatile therapeutic approach for diverse cancer types.

## 4. Discussion

Here, we report the anti-tumor effects of a diet containing the exogenous ketone ester, BD-AcAc_2_ in the context of standard percentages of carbohydrates, fats, and protein. We refer to this as the “eKET” diet, to distinguish it from KDs that typically contain < 5–10% of calories derived from carbohydrates and also contain high levels of fat and protein, which lead to the generation of endogenous KEs as a result of fat metabolism in the liver. Although there is growing interest in using both KDs and exogenous KEs as tools in the anti-cancer arsenal, particularly in the context of standard chemo- or immunotherapy use [6,18,36], few studies have examined the ability of the latter to slow the progression and dissemination of cancer cells in spontaneously metastasizing murine models. We chose to investigate the anti-cancer effects of our eKET diet in two orthotopic murine models of spontaneously metastasizing cancer: the 4T1 triple-negative breast cancer model and the Renca renal cancer model. The use of the eKET diet was able to significantly slow primary tumor growth and the formation of lung metastases in both models—without the addition of any standard cancer therapies. The eKET diet was found to be safe for use in the setting of disseminated cancers, as indicated by retained body mass at day 20 post-tumor challenge and endpoint weights of key organs, such as uninvolved kidneys. However, there were some indications of negative effects in the context of orthotopic and spontaneously metastasizing renal cancer, such as a significant loss of fat mass and liver weight, which should be investigated further to better understand translational implications. In addition, we found that eKET diet-induced gene expression patterns were markedly different in mammary versus renal tumors, indicating that the mechanisms responsible for controlling tumor progression varied in response to interactions between KEs and tumor-specific characteristics that have yet to be identified. Overall, however, this dietary approach has demonstrated safety and efficacy in preclinical models of two highly prevalent solid tumor types.

The rapid proliferation rate of many cancer subtypes means that tumors have high bioenergetic needs that are often met through glycolysis due to the Warburg effect, a metabolic state where cells shift from oxidative phosphorylation to rely more on the less efficient process of glycolysis to fulfill energy needs [37]. For this reason, nutritional interventions have been studied for their ability to disrupt the tumor’s energy supply and create a metabolic environment that inhibits cancer progression by limiting glycolysis, with a certain amount of success reviewed in [1,38]. One such approach is the KD, initially popularized for its positive effect on energy metabolism and weight loss [5]. During KD consumption, dietary and adipose-derived fatty acids exceed activation rates for key enzymes in the mitochondrial tricarboxylic acid (TCA) cycle in hepatocytes, leading to the production of water-soluble metabolites known as ketones, including AcAc and βHB. Ketones are transported from hepatocytes to extrahepatic tissues where they serve several important nutrient signaling properties and can also be fully oxidized in the TCA cycle for ATP production in the electron transport chain. In addition to their roles in nutrient signaling and energy metabolism, ketones have been studied by many groups for their intrinsic anti-tumor activities such as reducing systemic glucose availability, inhibiting histone deacetylases [6], reducing inflammation [8], impairing matrix metalloproteinase activity [9], and enhancing T cell immunity [6,10,18]. Therefore, the demonstrated metabolic and anti-tumor properties of ketones have increased interest in their use in the cancer setting.

Most prior reports on using KE supplementation to slow tumor growth have focused primarily on the inhibition of primary tumor progression and underlying mechanisms. For example, Murphy et al. used subcutaneous implantation of immune checkpoint blockade (ICB)-resistant prostate cancer cells to demonstrate that cyclic administration of a KD alone was nearly as effective as administered immunotherapy in slowing primary tumor outgrowth [6]. Notably, dietary administration of the same ketone ester used in our eKET diet formulation (BD-AcAc_2_) was also effective at delaying tumor outgrowth alone and in the context of ICB, through both tumor intrinsic mechanisms that included HDAC inhibition and extrinsic mechanisms that relied upon CXCL10/CXCR3-mediated recruitment of CD8+ T cells into tumors [6]. Metastatic disease was not evaluated in this model. A paper by Ferrere et al. also used both a KD and exogenous administration of either oral or intraperitoneal R-βHB to evaluate effects on tumor outgrowth and ICB efficacy [18]. Using the same Renca-Luc model we employed, this group found that KD reduced the rate of tumor expansion between days 7 and 15 post-tumor challenge—a finding replicated and expanded upon by our study (Figure 2). Lung metastases were not examined by Ferrere et al. Instead, the authors focused on deciphering the mechanisms by which KD or administered βHB slowed tumor outgrowth in a subcutaneous melanoma model, finding that oral βHB + ICB yielded the best survival results in a manner that appeared to depend upon beneficial gut microbiota changes and CXCR3 expression on CD8+ T cells [18]. Earlier work by Poff et al. examined the effects of exogenous KE supplementation on tumor progression in a model of disseminated glioblastoma following abdominal tumor challenge [17]. The authors found that dietary administration of the BD-AcAc_2_ resulted in a trending but non-significant reduction in tumor establishment in the brain, lungs, and liver. In another glioblastoma-focused paper, Lussier et al. found that administration of a KD increased CD4+ T cell infiltration within intracranial tumors and reduced the percentages of CD8+ T cells expressing PD-1 or CTLA-4, observations that were associated with increased CD8+ T cell cytolytic activity [39]. Together these studies demonstrate the ability of KE supplementation and/or KD use to slow primary tumor growth in a manner that involves enhanced anti-tumor immunity. Although important to our understanding regarding the potential for KE supplementation to augment ICB efficacy, these studies provide less information about other factors that are essential for a clinical translation of this approach, namely the effects of exogenous KEs on (1) spontaneous metastatic dissemination from orthotopic primary tumors; (2) broader transcriptional remodeling of the tumor microenvironment; and (3) animal health consequences that might limit clinical use. We have addressed each of these in our study.

Our findings demonstrate that the eKET diet reduces metastatic dissemination from established primary tumors. We reported previously that following intra-renal Renca-Luc tumor challenge, micrometastatic lesions are detectable in the lungs of 100% of mice by Day 7 using bioluminescent imaging [21]. The 4T1-Luc tumors also spontaneously metastasize to the lungs with 100% penetrance and are detectable using the same method [20]. Thus, both models are exceptionally well-suited for evaluating the effects of the eKET diet on the establishment of lung metastases. We found here that in mice with 4T1 tumors, the eKET diet significantly reduced lung tumor burdens by an average of 47% compared to mice on the CON diet (Figure 2). In contrast, although the eKET diet reduced lung tumor burdens by 96% in mice with Renca-Luc tumors, this reduction did not reach statistical significance, due to greater observed variability (Figure 2). Thus, dietary administration of eKET alone was able to produce substantial reductions in spontaneous metastases in two different pre-clinical tumor models. Although we did not examine whether ketone ester administration could directly limit tumor cell migration or invasion in vitro, future studies should examine these possibilities. These findings support future translational use of our eKET diet approach in patients with disseminated cancer, as an adjuvant to standard therapies.

Our NanoString gene expression analysis of primary tumors (Figure 5) provided mechanistic clues regarding the reduced development of lung metastases that could be pursued in future studies. In mice with 4T1 tumors, we found significant reductions in both *Wnt 4* and *Wnt10A*, genes encoding ligands that can activate downstream Wnt signaling, which has been linked to increased breast cancer metastasis [40]. We also found reductions in *Zeb1*, a master regulator of epithelial-to-mesenchymal transition (EMT) [41], which further suggests that the eKET diet may hinder invasive and migratory phenotypes, thereby suppressing metastasis. In addition, the downregulation of Ncam1 (neural cell adhesion molecule 1) suggests that the eKET diet may impair cell–cell adhesion and communication, potentially disrupting the metastatic cascade, and perhaps enhancing cancer cell killing by Natural Killer (NK) cells. Somewhat surprisingly, these same genes and pathways were not significantly downregulated in Renca-Luc tumors from mice on the eKET diet. Instead, we found a much less robust remodeling of the tumor microenvironment that was characterized by a distinct complement of genes that showed significant changes only in unadjusted *p*-values, which we used for screening purposes to obtain insight into underlying mechanisms. For example, the suppression of *Adm* (adrenomedullin), a potent angiogenic factor, aligns with previous studies demonstrating the antiangiogenic potential of ketogenic diet [42]. Inhibition of DNA repair (*Fanca*) and cell cycle progression (*Wdr76*) may sensitize Renca-Luc cells to DNA damage and apoptosis, further hindering tumor growth and survival. Similarly, the downregulation of *Hif1a*, a master regulator of the hypoxic response, suggests that our eKET diet may disrupt the adaptive mechanisms that allow tumors to thrive in low-oxygen environments, a hallmark of aggressive renal cancers [43,44,45].

Intratumoral hypoxia has been linked previously to accelerated renal cell carcinoma (RCC) progression but the regulation of and response to hypoxia in RCC is complex. Hypoxia has been found to induce expression of the protein PLOD2 in RCC via transcriptional activation by HIF1a; leading to downstream AKT activation and tumor progression [43]. Using a novel hypoxia-risk model, another group reported success in predicting outcomes of immune checkpoint blockade-based therapies [45]. This model was based on a hypoxia-related gene signature from Gene Set Enrichment Analysis (GSEA), which was then used to stratify RCC outcome data from TCGA; doing so illustrated that patients with a high-risk hypoxia gene expression score had worse OS than patients with a low-risk hypoxia score—findings that were validated in a subsequent cohort of 91 RCC patients from the International Cancer Genome Consortium database [45]. However, HIF1α is not always linked to tumor progression. Other studies indicate that HIF-1α acts as a tumor suppressor in clear cell RCC, with only HIF-2α exerting oncogene and pro-tumorigenic functions [44]. In support of this idea, another group found that in tumors from RCC patients, high *HIF1a* is associated with longer OS [46]. It is important to note here that the Renca cell line we used for our studies has wild type VHL expression and, therefore, relatively low endogenous levels of HIF1α; forced VHL knockdown in the Renca cell line causes high expression of HIF1α and accelerated tumor growth marked by phenotypic and morphologic changes consistent with epithelial-to-mesenchymal transition and increased metastases [47]. Thus, the reduction in both primary tumor growth and spontaneous lung metastases we observed with eKET diet administration (Figure 2)—and their association with reduced *Hif1a* gene expression (Figure 5)—are consistent with prior findings from the Wu lab [47].

It is essential for pre-clinical studies on potential anti-cancer agents to address the potential limitations of their use. Here, we observed distinct effects of our eKET diet in mice with mammary versus renal tumors, some of which suggest that the diet may be more difficult to translate into use for patients with metastatic RCC. Although mice with metastatic mammary tumors maintained body weight (Figure 4A), mice with renal tumors experienced a trending reduction in overall body weight (Figure 4D) that was accompanied by significant reductions in fat mass (Figure 4E) and liver weights (Figure 2B) at the experiment endpoint. These results suggest a negative interaction between eKET diet use and renal tumors that needs to be investigated further. At this time, the mechanism(s) responsible for these reductions have not been identified. It is important to note that mice with renal tumors on the eKET diet did not experience significant decreases in overall body weights or lean mass by QMR analysis and did not display atrophy of the calf muscle (excised gastrocnemius plus soleus) or tumor-free contralateral kidneys relative to mice on the CON diet. Thus, we conclude that the eKET diet is safe for use in both our breast cancer and renal cancer models, although there may be limitations regarding the duration of eKET diet use in the setting of patients with renal cancer.

## 5. Conclusions

Here, we have evaluated the ability of the exogenous ketone ester BD-AcAc_2_ for its anti-tumor and anti-metastatic effects in two different pre-clinical models of spontaneously metastasizing cancers. Both breast and renal cancers have been found previously to be highly glycolytic and lacking in their ability to metabolize ketones as an energy source [20]. For this reason, we selected cell lines representing these cancers for testing in syngeneic mice with intact immune systems: the mammary carcinoma line 4T-1 and the renal carcinoma line Renca. After confirming that our cell lines did indeed display robust glycolysis, we tested the effects of ketone ester administration both in vitro and in vivo. Although no direct inhibition of cancer cell proliferation was observed in vitro, we found substantial reductions in primary tumor outgrowth and metastatic dissemination in both models. Although we observed some negative effects of the eKET diet in the Renca renal cancer model that were not present in the 4T1 breast cancer model, we found our approach to be safe for use overall in the context of aggressive cancers. Our results suggest that a similar approach could be translated into clinical use as an adjuvant to standard cancer therapies. Additional research into the translational potential of the eKET diet is warranted.

## Figures and Tables

**Figure 1 cancers-16-03390-f001:**
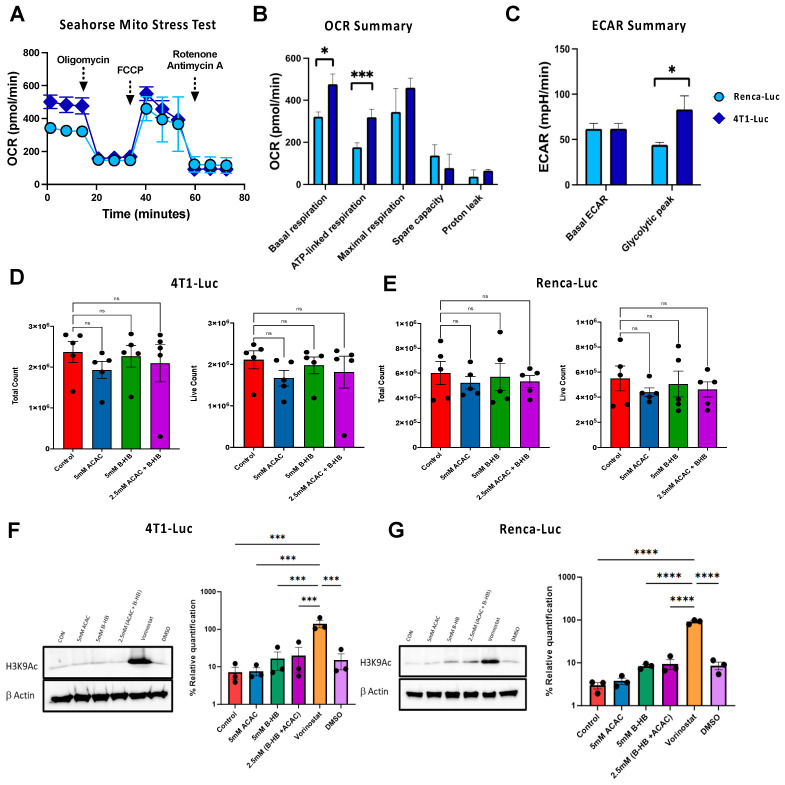
Metabolic and Cellular Phenotypes of 4T1 and Renca Cells and the Effects of Purified Ketone Esters. (**A**–**C**) Cellular metabolic characterizations via Seahorse (**A**) show significant differences in basal and ATP-linked respiration as per oxygen consumption rate (OCR) (**B**) and glycolytic peak as per extracellular acidification rate (ECAR) (**C**) in 4T1 versus Renca cells. (**D**,**E**) Purified KEs do not alter total and viable cell counts for either cell line during a 72-h incubation at the indicated concentrations. (**F**,**G**) Western blot and densitometric analysis of immunoblots of 4T1-Luc cells (**F**) or Renca-Luc cells (**G**) cultured for 24 h with purified ketone esters or Vorinostat. The HDAC inhibitor Vorinostat was used as a positive control to assess acetylated histone H3K9 (H3K9Ac). Both representative western blots and quantified data graphs are shown, with normalization to β-Actin control. Data are pooled from independent experiments and presented as means ± SEM. Statistical differences were calculated using parametric one-way ANOVAs or repeated measures two-way ANOVA with Tukey’s multiple comparisons tests for both (ns = not significant, * *p* < 0.05, *** *p* < 0.001, **** *p* < 0.0001).

**Figure 2 cancers-16-03390-f002:**
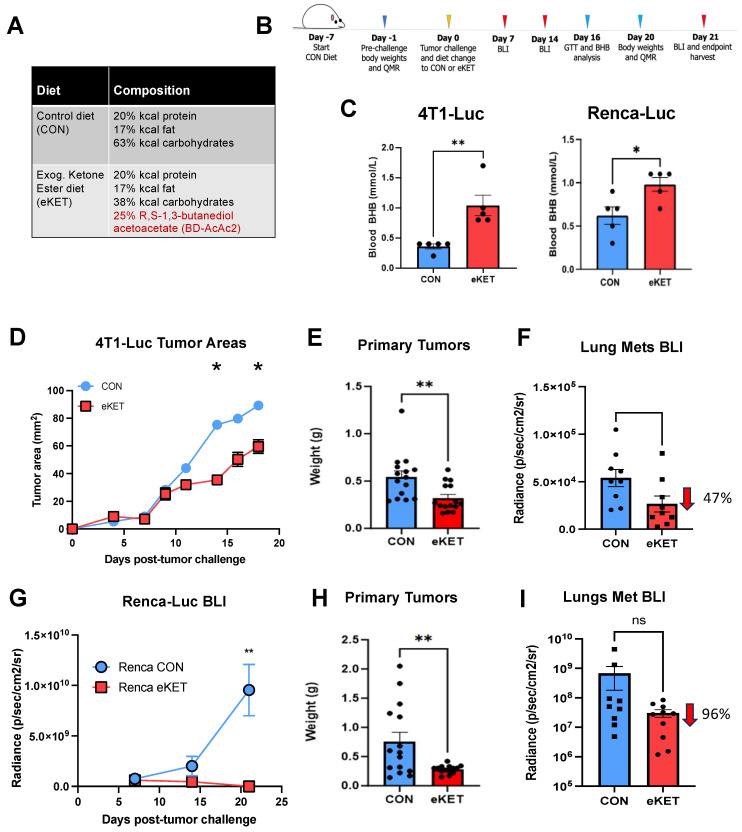
Administration of an eKET diet slows primary tumor progression and reduces spontaneous lung metastases in both 4T1-Luc mammary and Renca-Luc renal tumor models. (**A**) Macronutrient composition (percentage of kilocalories) of the control diet (CON) and the exogenous ketone ester diet (eKET). (**B**) Schematic representation of the experimental design for panels (**C**–**I**). (**C**) Serum b-hydroxybutyrate (BHB) levels in BALB/c mice bearing 4T1-Luc mammary carcinomas or Renca-Luc renal tumors fed either the CON or eKET diet, measured on Day 16 post-challenge. (**D**) Primary mammary tumor burdens over time in 4T1-Luc tumor-bearing mice measured by vernier calipers. (**E**) Excised mammary carcinoma tumor weights of 4T1-Luc tumor-bearing mice on day 21 post-tumor challenge. (**F**) Lung metastases in 4T1-Luc tumor-bearing mice measured by bioluminescent imaging (BLI) on Day 21 post-tumor challenge. (**G**) Primary renal tumor burdens over time in Renca-Luc tumor-bearing mice measured by BLI. (**H**) Excised renal carcinoma tumor weights of Renca-Luc tumor-bearing mice on day 21. (**I**) Lung metastases in Renca-Luc tumor-bearing mice measured by BLI on Day 21 post-tumor challenge. Data represent mean ± SEM. Statistical significance was assessed using unpaired *t*-tests or Mann–Whitney U tests for two-group comparisons, repeated measures two-way ANOVA with post-hoc tests for time-course data with Tukey’s multiple comparisons test Significance levels: * *p* < 0.05, ** *p* < 0.01; ns, not significant.

**Figure 3 cancers-16-03390-f003:**
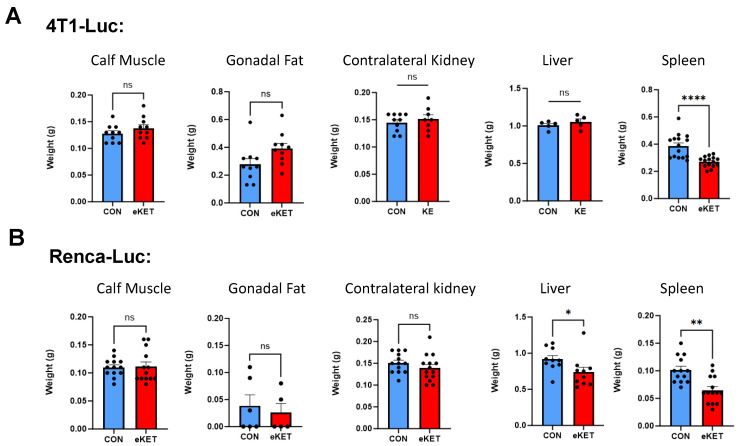
The eKET diet maintains key muscle and organ weights in mice with 4T1-Luc mammary tumors and abates splenomegaly but causes undesirable reductions in liver weights in mice with Renca-Luc tumors. (**A**) Calf, Gonadal fat, Contralateral kidney, Liver weight in 4T1-Luc tumor-bearing mice fed with Control vs. eKET diet at Day 21 post-tumor implant. (**B**) Calf, Gonadal fat, Contralateral kidney, Liver weight Renca-Luc tumor-bearing mice fed with Control vs. eKET diet at Day 21 post-tumor implant. Data are presented as the means ± SEM. Statistical differences were determined by parametric *t*-tests or nonparametric Mann–Whitney tests as appropriate with Tukey’s post-hoc test (ns = not significant; * *p* < 0.05; ** *p* < 0.01; **** *p* < 0.0001). CON = control diet; eKET = exogenous ketone ester diet.

**Figure 4 cancers-16-03390-f004:**
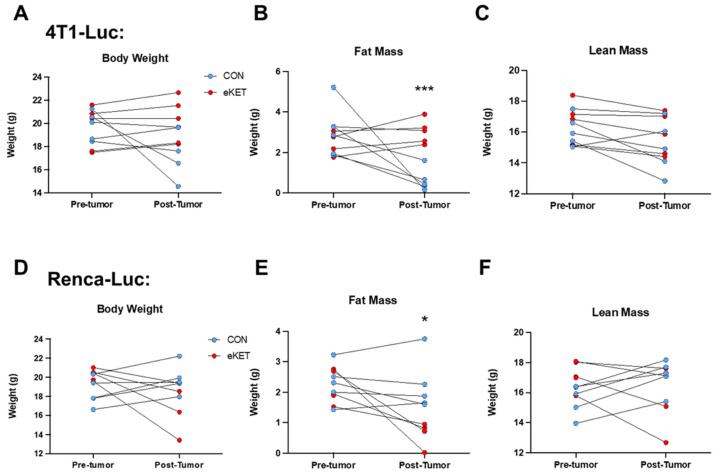
Administration of an exogenous Ketone Ester (eKET) diet after tumor challenge has variable effects on body composition. (**A**) Body weights of individual 4T1-Luc tumor-bearing mice (*n* = 5 each CON and eKET diet) at Day 1 and Day 20, relative to tumor challenge. Changes in (**B**) fat mass or (**C**) lean mass for the same 4T1 tumor-bearing mice are shown in (**A**). (**D**) Body weights of individual Renca-Luc tumor-bearing mice (*n* = 5 each CON and eKET diet) at Day 1 and Day 20, relative to tumor challenge. Changes in (**E**) fat mass or (**F**) lean mass for the same Renca-Luc tumor-bearing mice shown in (**D**). Data represent mean ± SEM. Statistical differences were determined by parametric *t*-tests or nonparametric Mann–Whitney tests as appropriate with Tukey’s post-hoc test (* *p* < 0.05; *** *p* < 0.001). CON = control diet; eKET = exogenous ketone ester diet.

**Figure 5 cancers-16-03390-f005:**
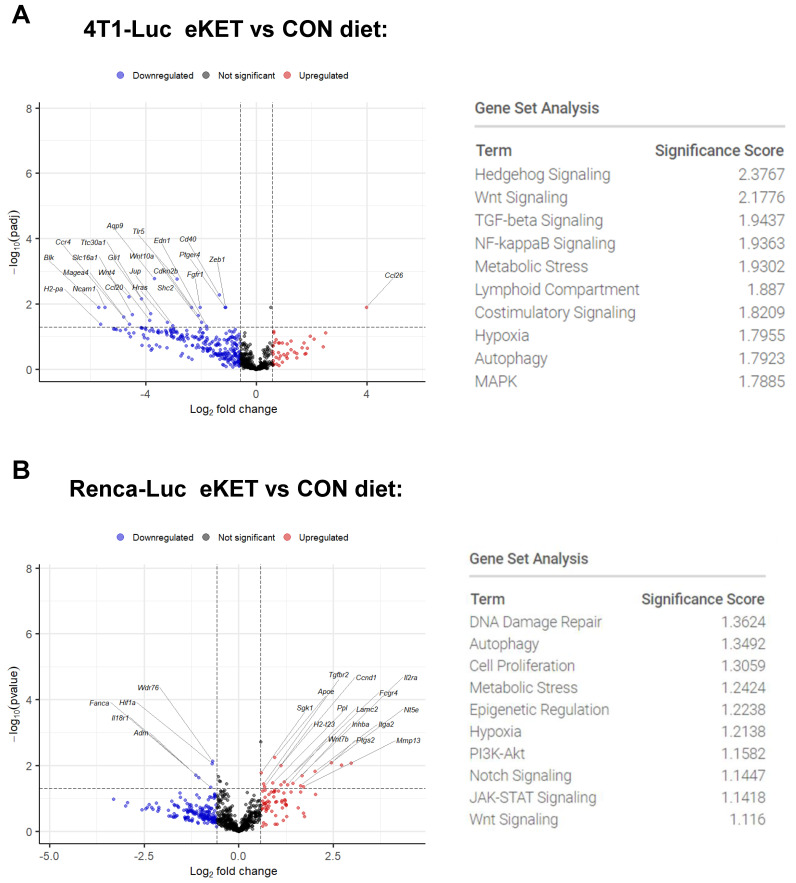
eKET diet-induced reductions in primary tumors are associated with distinct changes in the tumor microenvironments of mammary carcinoma and renal tumors. (**A**,**B**) Volcano plots showing differential gene expression in (**A**) 4T1-Luc and (**B**) and Renca-Luc tumor-bearing mice fed either a control (CON) or eKET (eKET) diet. Each dot represents a gene, with the *x*-axis indicating the log2 fold change in expression between the two diet groups and the *y*-axis indicating the −log10 of the *p*-adj value for (**A**) and the −log10 of the *p*-value for (**B**). Significantly upregulated genes are shown in red, significantly downregulated genes are shown in blue, and non-significant genes are shown in black. The top 10 most significant pathways affected by the eKET diet in each tumor model are listed in the tables next to the volcano plots, along with their significance scores.

## Data Availability

The raw data supporting the conclusions of this article will be made available by the authors on request.

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
