# Peer review of "An Exogenous Ketone Ester Slows Tumor Progression in Murine Breast and Renal Cancer Models"

_cancers, 2024, doi:10.3390/cancers16193390_

Round 1

Reviewer 1 Report

Comments and Suggestions for Authors

The manuscript by Ogbonna and colleagues describes the effect of feeding ketone esters to ectopic mice tumor models. Initially luciferase transfected mouse mammary tumor or renal tumor cells were analyzed for metabolic adaptations. No differences in cell proliferation with ketone ester treatments were observed in vitro. Cells were then injected into BALB/c mice and mice were fed Acetoacetate Esters to replace about 25% of calories. Primary tumors size as well as lung metastases decreased for both models. Splenomegaly was reduced in both models. Fat mass significantly decreased in the renal tumor model with ketone ester supplementation while it significantly decreased with the control diet in the mammary cancer model. Differential gene expression shows a pronounced downregulation in the mammary tumor models.  Overall, the authors conclude that ketone ester supplementation without ketogenic decrease in carbohydrate intake has similar effects as ketogenic diets alone but may be more palatable to the patient population.

Overall, the manuscript is well written and presents a significant body of work. The experiments seem to mostly be well conducted, and the statistical analyses are appropriate. However there are certain analyses that seem to not quite fit the flow of the argument. I will start with those first:

The metabolomic characterization of the cell lines includes OC and ECAR measurements and cell proliferation, but the inclusion of a specific acetylated histone doesn’t fit, particularly since it doesn’t show any significant change. A more appropriate characterization to include here might be migration and invasion assays. This is particularly appropriate since the premise of the paper is to specifically investigate metastasis as stated in the abstract, while no there is no mention of any interest in epigenetic modifications.

The second experiment that doesn’t quite fit and may need further explanation is the GTT halfway through the experimentation. I wanted to comment on how do the authors now that it is the supplementation of KE that has the described effects and not a potential reduction in blood glucose levels, given that the ketogenic diet is thought to decrease tumor proliferation at least in part by decreasing available glucose. So I was expecting glucose levels to have been measured throughout the 21 day experiment, but instead the authors performed a GTT? It would be interesting to have some more information on the appropriateness of this experiment, particularly right in the middle of KE supplementation period.

This brings me to another point. It does not seem that the authors have used control mice without injected tumors. Since the authors do present a number of physiological changes, such as key muscle and organ weights as well as body composition. It would be great to compare these findings to non-tumor carrying mice. The difference between the two tumor models is certainly of interest, but the effect of KE supplementation on healthy mice would have provided an additional datapoint that would be crucial to determine any tumor independent effects of KE supplementation.

Lastly, the mice are fed with Acetoacetate supplements, but the blood ketone measures beta-hydroxybutyrate (BHB) concentrations. While I agree that the main blood ketone will be BHB, it seems odd to assume that liver and peripheral tissue would go through the NADPH-dependent process of reducing BHB out or circulating Acetoacetate. Is it therefore at least a possibility that the authors are measuring endogenous BHB production rather than their exogenous Acetoacetate? Even the authors seem to indicate that the mice are in mild ketosis line 299.

Author Response

Reviewer #1

The manuscript by Ogbonna and colleagues describes the effect of feeding ketone esters to ectopic mice tumor models. Initially luciferase transfected mouse mammary tumor or renal tumor cells were analyzed for metabolic adaptations. No differences in cell proliferation with ketone ester treatments were observed in vitro. Cells were then injected into BALB/c mice and mice were fed Acetoacetate Esters to replace about 25% of calories. Primary tumors size as well as lung metastases decreased for both models. Splenomegaly was reduced in both models. Fat mass significantly decreased in the renal tumor model with ketone ester supplementation while it significantly decreased with the control diet in the mammary cancer model. Differential gene expression shows a pronounced downregulation in the mammary tumor models.  Overall, the authors conclude that ketone ester supplementation without ketogenic decrease in carbohydrate intake has similar effects as ketogenic diets alone but may be more palatable to the patient population.

Overall, the manuscript is well written and presents a significant body of work. The experiments seem to mostly be well conducted, and the statistical analyses are appropriate. However, there are certain analyses that seem to not quite fit the flow of the argument. I will start with those first:

The metabolomic characterization of the cell lines includes OC and ECAR measurements and cell proliferation, but the inclusion of a specific acetylated histone doesn’t fit, particularly since it doesn’t show any significant change. A more appropriate characterization to include here might be migration and invasion assays. This is particularly appropriate since the premise of the paper is to specifically investigate metastasis as stated in the abstract, while no there is no mention of any interest in epigenetic modifications.

Author response: Thank you for your valuable feedback. While we did not observe significant changes in H3K9Ac levels following purified KE treatment, this finding is still scientifically important. Our aim was to explore whether KEs exert any epigenetic effects, particularly through histone acetylation, which is known to influence cancer cell behavior (growth and proliferation). The lack of significant changes in H3K9Ac suggests that the anti-tumor effects of KEs may operate independently of this specific epigenetic mechanism. We agree that in vitro migration and invasion assays could provide additional insights, especially given the focus on growth and metastasis outlined in our abstract. However, as noted, we have demonstrated in vivo that ketone diesters, when formulated as part of the diet, can effectively prevent tumor growth and metastasis. This finding addresses the core focus of our study. For future studies, we plan to expand our investigation to include migration and invasion assays, which are relevant to our findings of reduced metastasis in vivo.

Action taken: We included a statement in the Discussion to highlight the utility of performing migration and invasion assays in the future: “Although we did not examine whether ketone ester administration could directly limit tumor cell migration or invasion in vitro, future studies should examine these possibilities.” (lines 567-569)

The second experiment that doesn’t quite fit and may need further explanation is the GTT halfway through the experimentation. I wanted to comment on how do the authors know that it is the supplementation of KE that has the described effects and not a potential reduction in blood glucose levels, given that the ketogenic diet is thought to decrease tumor proliferation at least in part by decreasing available glucose. So I was expecting glucose levels to have been measured throughout the 21 day experiment, but instead the authors performed a GTT? It would be interesting to have some more information on the appropriateness of this experiment, particularly right in the middle of KE supplementation period.

Author response: Thank you for this important comment. We now include data at day 16 post-tumor challenge - a time when both tumor types are growing robustly in mice on the CON diet – to show that no significant differences in fasting blood glucose are present in mice on eKET versus CON diet (revised Supplemental Figure 1). Regarding the GTT, our rationale for conducting this assay is based on the fact that the ketone diester is administered as part of the diet, in a carbohydrate replete state (albeit with a 25% reduction in carbohydrate energy to accommodate the ketone ester with the purpose of creating an isocaloric diet). Thus, dietary ketones reach the circulation in the presence of glucose. A reduction in circulating glucose concentrations during the GTT was deemed to be more reflective of the day-to-day consumption of the two diets for comparison purposes since a lower circulating glucose concentration during the GTT is a reflection of the well-known reductions in circulating glucose that occur following ingestion of ketone esters.

Action taken: We have revised Supplemental Figure 1 to show data from fasted blood glucose testing at day 16 post-tumor challenge and have added the following to the Results section: “Simultaneous fasted blood glucose concentrations revealed no significant alterations between mice on CON versus eKET diets (Supplemental Figure 1).” (lines 324-325)

This brings me to another point. It does not seem that the authors have used control mice without injected tumors. Since the authors do present a number of physiological changes, such as key muscle and organ weights as well as body composition. It would be great to compare these findings to non-tumor carrying mice. The difference between the two tumor models is certainly of interest, but the effect of KE supplementation on healthy mice would have provided an additional datapoint that would be crucial to determine any tumor independent effects of KE supplementation.

Author response: This is a valid concern and one we shared with the reviewer, despite our group’s prior careful evaluations of physiologic and metabolic responses to the ketone diester used in this study in juvenile and adult Bl/6 mice on various dietary and environmental backgrounds. Our published findings consistently demonstrate in lean mice on a low-fat diet or a high-fat high-sugar diet that the ketone diester attenuates the accretion of body weight, which is accounted for with near entirety by reductions in adiposity and overall maintenance of lean body mass. We have fully characterized the responses and find that energy intake is similar between control and ketone diester treated mice, whereas markers of energy expenditure and energy loss through feces and urine may contribute to the phenotype in mice without cancer. Please find a listing of those papers at the following link: Plaisance ep ketone esters - Search Results - PubMed (nih.gov).

To address the possible independent effects of KE supplementation on tumor-free BALB/c mice, we randomized mice to CON or eKET diet and housed them alongside tumor-challenged mice in our animal facility for the duration of our tumor-challenge period. We then euthanized the tumor-free mice and weighed vital organs, including the liver, kidneys, spleens, and gonadal fat pads. These data are included in Supplementary Figure 3. As shown, the lack of an impact of KEs on muscle and organ weights in healthy mice provides an important reference point for understanding the metabolic shifts induced by KE treatment. This comparison reinforces our conclusion that the observed physiological changes in tumor-bearing mice are largely a consequence of the tumor microenvironment and the metabolic alterations associated with cancer progression, rather than KE treatment alone.

Action taken: We revised the Results text to emphasize pre-existing data found in Supplemental Figure 3, showing vital organ and muscle weights in tumor-free mice randomized to eKET versus CON diets and housed alongside our tumor-challenged mice for the duration of our 21-day tumor growth period. “In agreement with these prior reports, we found that in tumor-free mice, eKET administration did not significantly alter excised calf muscle, liver, spleen, or contralateral kidney weights after three weeks of diet administration, indicating that the diet is well-tolerated and does not induce adverse systemic effects in healthy animals (Supplemental Figure 3).” (lines 375-377)

Lastly, the mice are fed with Acetoacetate supplements, but the blood ketone measures beta-hydroxybutyrate (BHB) concentrations. While I agree that the main blood ketone will be BHB, it seems odd to assume that liver and peripheral tissue would go through the NADPH-dependent process of reducing BHB out or circulating Acetoacetate. Is it therefore at least a possibility that the authors are measuring endogenous BHB production rather than their exogenous Acetoacetate? Even the authors seem to indicate that the mice are in mild ketosis line 299.

Author Response: The ketone diester is R,S-1,3-butanediol diacetoacetate. While the reviewer is correct that 50% of the molecule is acetoacetate, the remaining 50% of the ketone ester is butanediol which is converted by hepatic aldehyde and alcohol dehydrogenases into BHB. Although it is not completely understood, it has been proposed that exogenous acetoacetate is taken up by extrahepatic tissues and then undergoes complete oxidation in the TCA cycle with the oxidation/reduction reaction catalyzed by the enzyme SCOT. Butanediol taken up by hepatocytes is converted directly to BHB and presumably is exported from the liver where it is then converted to acetoacetate in extrahepatic tissues by the enzyme BDH1 which then readily provides additional reducing equivalents for the TCA cycle. From a redox perspective, we acknowledge that some acetoacetate may be taken up by the liver and converted to BHB, but again this has not been well characterized in the field.

The main limitation for the field is that point of care ketone device technology does not measure circulating acetoacetate concentrations and in the current study we did not collect samples to run acetoacetate on LC/MS platforms that are typically used to evaluate circulating concentrations of acetoacetate.

The other aspect for consideration by the reviewer is that while circulating BHB or AcAc concentrations are important to determine whether tissues are seeing the ketones administered, it does not address the more important issue of clearance or tissue concentrations of the ketones.

In summary, our purpose by providing the BHB is warranted because the ester does in fact get reduced to BHB and also because our goal was simply to demonstrate that the ketone diester is indeed increasing circulating concentrations, with the limitation that this routinely used method provides nothing more than a surrogate marker that uptake has occurred.

Action taken: Text in the Methods section was revised to clarify that point of care ketone device technology does not permit the evaluation of circulating acetoacetate: “Because point of care ketone device technology is not capable of measuring circulating acetoacetate concentrations, only concurrent blood b-hydroxybutyrate (β-HB) concentrations were measured.” (lines 181-183). Text in the Results was revised to emphasize that the ketone diester incorporated into our eKET diet is a precursor to both β-HB and AcAc: “…wherein 25% of the carbohydrate-derived calories were replaced with BD-AcAc2 (Figure 2A), a ketone diester precursor to R-β-HB and AcAc…” (lines 314-315)

Reviewer 2 Report

Comments and Suggestions for Authors

The manuscript “An exogenous ketone ester slows tumor progression in murine breast and renal cancer models” presents an interesting topic and it is suitable for the publication in the journal. The manuscript has been reviewed. The presented work requires minor modifications.  My specific comments are as follows:

1.       The manuscript quantity and quality are sufficient for publication in the journal. The manuscript has been well organized. The language is good and does not contain grammatical errors.

2.       The data presented is sufficient to explain the findings.

3.       The introduction should be added with adverse effects associated with the administration of the keto diet. In addition, its interaction with other drugs and formulations should be addressed briefly.

4.       Add the Animal Ethical approval number obtained to carry out the study.

5.       Although ketogenic diet has advantages. But it also led to side effects such as risk of heart disease etc. Does these factors were considered in the experimental plan? Please include the relevant factors and justify how the current research correlates with these morbidities?

6.       Is there any specific reason to choose only breast and renal cancers? Keto diets are known to cure neurological disease effectively. Should you choose the brain or related tumors?

7.       A few studies were conducted along with the hemotherapy: Yang, Lifeng, et al. "Ketogenic diet and chemotherapy combine to disrupt pancreatic cancer metabolism and growth." Med 3.2 (2022): 119-136 (not the reviewer work). Please discuss such works with your findings.

Author Response

The manuscript “An exogenous ketone ester slows tumor progression in murine breast and renal cancer models” presents an interesting topic and it is suitable for the publication in the journal. The manuscript has been reviewed. The presented work requires minor modifications.  My specific comments are as follows:

  1. The manuscript quantity and quality are sufficient for publication in the journal. The manuscript has been well organized. The language is good and does not contain grammatical errors. Thank you.
  2. The data presented is sufficient to explain the findings. Thank you.
  3. The introduction should be added with adverse effects associated with the administration of the keto diet. In addition, its interaction with other drugs and formulations should be addressed briefly. 

Author Response: The ketone diester used in this study was administered under a carbohydrate replete dietary background with only 25% of energy from carbohydrate removed to create isocaloric control and eKET diets. We are aware of the ketogenic diet literature and are aware that there are very inconsistent findings, particularly in the human literature, with regard to the anti-tumorigenic aspects of ketogenic diets. These prior inconsistencies with ketogenic diet administration drove our decision to test a fundamentally different approach. We ask the reviewer to note that our approach of exogenous ketone administration under a carbohydrate replete state is distinct from that of a ketogenic diet where carbohydrates are essentially depleted. To conclude, this was not a ketogenic diet study and thus we would prefer to not present any discussion related to side effects associated with ketogenic diet, so as not to confuse readers by emphasizing an approach we did not utilize.

  1. Add the Animal Ethical approval number obtained to carry out the study. This has been added to the Methods section on line #126.
  2. Although ketogenic diet has advantages. But it also led to side effects such as risk of heart disease etc. Does these factors were considered in the experimental plan? Please include the relevant factors and justify how the current research correlates with these morbidities?

Author Response: Again, please note that this was not a ketogenic diet study and thus we would prefer to not present any discussion related to side effects associated with ketogenic diet, so as not to confuse readers by emphasizing an approach we did not utilize.

Action taken: We have revised the Methods text to clarify that our eKET diet is NOT a ketogenic diet: “Therefore, the eKET Diet is a carbohydrate-replete diet with normal percentages of fat and protein, which distinguishes it from “ketogenic” diets.” (lines 121-123) We also modified the Results text: “mice were randomized to either continue on CON diet or switch to the eKET diet (Figure 2A and B), a carbohydrate-replete diet…” (lines 313-314)

  1. Is there any specific reason to choose only breast and renal cancers? Keto diets are known to cure neurological disease effectively. Should you choose the brain or related tumors?

Author response: Thank you for your insightful comment. We chose breast and renal cancers for this study based on the findings from Qian et al., 2023 whose analysis of over 10,000 tumor specimens in The Cancer Genome Atlas (TCGA) revealed that both breast cancer and renal carcinoma are characterized by high glycolytic activity and diminished ketone ester metabolism. This metabolic profile aligns with our focus on understanding the therapeutic potential of ketogenic interventions in cancer models where glycolysis is a dominant metabolic pathway. While ketogenic diets are indeed effective in neurological diseases, particularly in managing certain brain tumors, our goal was to target cancers that exhibit significant alterations in glycolytic and ketone metabolism. Therefore, breast and renal cancer models were selected as they provide a relevant context for investigating these metabolic shifts. We appreciate your suggestion, and it presents an interesting opportunity for future research, especially regarding brain tumors and related malignancies.

  1. A few studies were conducted along with the hemotherapy: Yang, Lifeng, et al. "Ketogenic diet and chemotherapy combine to disrupt pancreatic cancer metabolism and growth." Med 3.2 (2022): 119-136 (not the reviewer work). Please discuss such works with your findings. 

Author Response: Thank you for your valuable comment and for bringing the work of Yang et al. (2022) to our attention. Their study provides valuable insights into the potential benefits of combining ketogenic diets with chemotherapy in cancer treatment. While our study aligns with the broader investigation of dietary interventions in cancer therapy, we would like to emphasize a key distinction between our approach and that of Yang et al. The ketone diester used in our study was administered under a carbohydrate-replete dietary background, with only 25% of energy from carbohydrates removed to create isocaloric control and eKET diets. This differs from the traditional ketogenic diet, which typically involves a more stringent restriction of carbohydrates. However, we think that future research to evaluate whether our exogenous ketogenic esters can work synergistically with chemotherapies in breast and renal cancers would be important.

Action Taken: Nevertheless, we have revised the Discussion text to cite this reference as an example of potential clinical applications of our approach: “Although there is growing interest in using both KDs and exogenous KEs as tools in the anti-cancer arsenal, particularly in the context of standard chemo- or immunotherapy use,1-3 few studies have examined the ability of the latter to slow the progression and dissemination of cancer cells in spontaneously metastasizing murine models.” (lines 481-483)

Reviewer 3 Report

Comments and Suggestions for Authors

The article entitled “An exogenous ketone ester slows tumor progression in murine breast and renal cancer models” examines the effects of the eKET diet in mouse models of different cancer (breast and kidney cancers). The experimental design is clearly reported. The manuscript would be of general interest to the researchers of this field. Before taking full acceptance, the authors should consider and incorporate in the present form of the manuscript. Here are some concerns that need to be addressed in the present form of the manuscript.

Some comments and corrections for authors:

1.     Overally, the manuscript has some punctuation and grammatical errors and needs to be corrected (i.e., there must be comma before and in all mns when mention about over two parameters). Please run throughout the mns.

2.     Abstract section before enlarging with some important biological results must be reorganized.

3.     The negative effects of eKET diet must be explained in details with observations in the experiments.

Comments on the Quality of English Language

Minor editing of English language required.

Author Response

Reviewer #3

The article entitled “An exogenous ketone ester slows tumor progression in murine breast and renal cancer models” examines the effects of the eKET diet in mouse models of different cancer (breast and kidney cancers). The experimental design is clearly reported. The manuscript would be of general interest to the researchers of this field. Before taking full acceptance, the authors should consider and incorporate in the present form of the manuscript. Here are some concerns that need to be addressed in the present form of the manuscript.

Some comments and corrections for authors:

  1. Overally, the manuscript has some punctuation and grammatical errors and needs to be corrected (i.e., there must be comma before and in all mns when mention about over two parameters). Please run throughout the mns.

Author Response: Thank you. We have performed a spelling and grammar check on our revised manuscript.

  1. Abstract section before enlarging with some important biological results must be reorganized.

Author Response: Thank you. We have revised the Abstract as suggested. (lines 34-44)

  1. The negative effects of eKET diet must be explained in details with observations in the experiments.

Author Response: The few negative effects of the eKET diet we observed were in fact summarized in the Abstract, detailed in the Results (see our original and un-revised section 3.4 entitled “The Safety Profile of eKET Diet Administration Differs Between Mammary and Renal Tumor Models”, mentioned in the Discussion (see However, there were some indications of negative effects in the context of orthotopic and spontaneously metastasizing renal cancer, such as a significant loss of fat mass and liver weights, which should be investigated further to better understand translational implications.” [lines 492-495] and “It is essential for pre-clinical studies on potential anti-cancer agents to address potential limitations of their use. Here, we observed distinct effects of our eKET diet in mice with mammary versus renal tumors, some of which suggest that the diet may be more difficult to translate into use for patients with metastatic RCC. Although mice with metastatic mammary tumors maintained body weight (Figure 4A), mice with renal tumors experienced a trending reduction in overall body weight (Figure 4D) that was accompanied by significant reductions in fat mass (Figure 4E) and liver weights (Figure 2B) at experiment endpoint. These results suggest a negative interaction between eKET diet use and renal tumors that needs to be investigated further.” [lines 625-626], and reiterated in the Conclusions section “Although we observed some negative effects of the eKET diet in the Renca renal cancer model that were not present in the 4T1 breast cancer model, we found our approach to be safe for use overall in the context of aggressive cancers.” [lines 647-649]

Action Taken: We respectfully believe these existing text passages regarding the negative effects of our eKET diet in the context of renal cancer are sufficient to address the Reviewer’s concern.